# Complement Properdin Determines Disease Activity in MRL/*lpr* Mice

**DOI:** 10.3390/medicina56090430

**Published:** 2020-08-27

**Authors:** Hasanain Alaridhee, Azzah Alharbi, Zeayd Saeed, Róisín C. Thomas, Cordula M. Stover

**Affiliations:** 1Department of Respiratory Sciences, University of Leicester, Leicester LE1 9HN, UK; hasanain.alarithy@gmail.com (H.A.); azzaal-harbi@hotmail.com (A.A.); zeaydfadhil@gmail.com (Z.S.); roisinthomas@gmail.com (R.C.T.); 2Department of Medical Microbiology and Parasitology, King Abdulaziz University, Jeddah 21589, Saudi Arabia; 3Department of Nursing, Technical Institute of Samawa, Al-Furat Al-Awsat Technical University, Najaf 54003, Iraq; 4Department of Cardiovascular Sciences, University of Leicester, Leicester LE1 9HN, UK

**Keywords:** lupus, properdin, mouse model

## Abstract

*Background and objects:* In systemic lupus erythematosus, circulating immune complexes activate complement and, when trapped in renal capillaries, cause glomerulonephritis. Mouse models have been used in the preclinical assessment of targeting complement activation pathways to manage chronic inflammation in lupus. Properdin is the only known positive regulator of complement activation, but its role in the severity of lupus nephritis has not been studied yet. *Materials and Methods:* Fully characterized properdin-deficient mice were crossed with lupus prone MRL/*lpr* mice on C57Bl/6 background. *Results:* Compared to MRL/*lpr* properdin wildtype mice, MRL/*lpr* properdin-deficient mice had significantly lower anti-DNA antibody titres, TNFα and BAFF levels in serum. The qualitative glomerulonephritic score was less severe and there was significantly less serum creatinine in MRL/*lpr* properdin-deficient mice compared to MRL/*lpr* properdin wildtype littermate mice. *Conclusion:* Properdin plays a significant role in the severity of lupus overall and specifically in the extent of glomerulonephritis observed in MRL/*lpr* mice. Because MRL/*lpr* properdin-deficient mice had lower levels of anti-DNA antibodies, inflammatory mediators and markers of renal impairment, the study implies that properdin could constitute a novel therapy target in lupus disease.

## 1. Introduction

The complement system, a cascade of carefully regulated enzymatic reactions, bridges innate and adaptive immunity. Three main pathways have been described, whose activity can be specifically interrogated in vitro: the classical pathway is activated when antigen–antibody complexes have formed; the lectin pathway is initiated upon the recognition of pathogen associated molecular patterns; the alternative pathway amplifies the activities of both the classical and lectin pathways, while it may become a dominant pathway itself when the host relies on the binding of properdin to, for example, certain serotypes of *Neisseria meningitidis*. Normally, complement activation is well controlled through the presence of regulators on the surface of host cells or in circulation [1]. Complement receptors mediate cellular effects, such as inflammatory cell recruitment, phagocytosis, and B cell activation. The non-enzymatic formation of the membrane attack complex leads to pore formation, causing the rapid lytic destruction of damaged cells (or some Gram-negative microorganisms) but is also detectable in a soluble form in systemic lupus erythematosus (SLE) [2].

In humans, the loss of self-tolerance leads to this disease, which varies in its manifestation and severity. It is a B-lymphocyte hyper-reactive disease leading to the production of autoantibodies [3]. SLE is a type-III hypersensitivity response with type-II involvement [4], meaning that the disease is characterized by circulating antigen–antibody complexes and the presence of cells tagged by bound antibodies. Both type II and type III responses entail the involvement of complement activation via the recognition of immune complexes using C1 of the classical pathway [5]. Elevated serum levels of activation products of complement are seen in patients with SLE [6].

SLE affects 1 in 2000 Europeans [7] but the prevalence is much higher in populations of other extractions [8], in whom severity is much worse [9]. Deposited immune complexes are pathogenic in developing vasculitis, arthritis, encephalitis, foetal loss, and nephritis. The impairment of kidney function is generally life limiting, requiring dialysis or renal transplant. Renally deposited immune complexes contribute to the development of glomerulonephritis, because of the local activation of complement that leads to the formation of the chemoattractants C3a and C5a, and membrane attack complexes. The excretion of C3d, an activation product of complement C3, in urine, may be a marker of severity for lupus nephritis [10].

Properdin is composed of thrombospondin-like repeats of which some have been identified to bind to C3b, which are present in the C3 convertase complex of the alternative pathway of complement activation (C3bBb) and in the C5 convertase complexes of the alternative pathway and classical/lectin pathways (C3bBbC3b, C4b2b3b). The binding of properdin to C3b in these complexes leads to a stabilisation of these inherently labile convertases. A binding independent of C3b to membranes via phosphatidylserine is described for properdin [11,12] and is relevant to SLE where increased apoptosis occurs. Properdin has been measured in patients with SLE and was found to be decreased [13], which was most likely due to consumption, a phenomenon routinely observed during the chronic activation of complement in SLE, when the capacity to remove immune complexes becomes exhausted.

The significance of alternative pathway amplification to complement activation initiated by immune complexes has been quantified in vitro [14] and demonstrated in vivo during antipneumococcal antibody response [15]. Deficiencies in humans predisposed to septicaemia with *Neisseria meningitidis* serotypes, W-135 and Y, but more recently, properdin synthesized by cells of the myeloid lineage, are valued as a modulators in the outcome of tissue injury [16].

The disease mechanisms of SLE can be modelled in mice. This study was particularly interested in the development and progression of glomerulonephritis. A strain of MRL/MpJ-Faslpr (MRL/*lpr*) mice was chosen that, spontaneously, as part of a lymphoproliferative disorder, develops systemic autoimmune disease affecting the kidney. MRL/*lpr* mice produce anti-DNA autoantibodies and renal immune deposits which contain IgG, IgM, IgA and C3. In the genetic absence of Factors B or D (proteases of the alternative pathway that aid in C3 activation), the lupus-like phenotype in MRL/*lpr* mice was significantly improved [17,18].

Therefore, the aim of this study was to determine whether, in the genetic absence of properdin, the systemic and organ specific inflammation was less severe, by using a novel mouse model generated by crossing the repository archived lupus-prone MRL/MpJ-Fas*lpr*/J mice with our line of properdin knock-out mice to produce a strain of lupus-prone properdin-deficient (MRL/*lpr* P^KO^) mice, and lupus-prone properdin-sufficient littermates (MRL/*lpr* P^WT^) as controls. A gene dose effect was studied in lupus-prone properdin-heterozygous mice (MRL/*lpr* P^het^). The results reported herein demonstrate that pathologic and functional renal disease were significantly reduced in MRL/*lpr* P^KO^ mice confirming that the alternative pathway amplification plays a significant role in the proliferative glomerulonephritis that develops in MRL/*lpr* mice.

## 2. Materials and Methods

### 2.1. Generation of Properdin Deficient Fas Mutant Mice

Since their generation, properdin deficient mice, in comparison with their congenic controls, were analysed across their systems [19]. In a functional test of serum obtained from properdin deficient mice, we showed that properdin was the dominant factor in rabbit red blood cell lysis in buffers favouring alternative pathway activation (without calcium ions) [20]. MRL/*lpr* mice (B6.MRL-Fas^lpr^/Orl) were obtained from the European Mouse Mutant Archive (EMMA) mouse repository (INFRAFRONTIER GmbH), rederived, then crossed with properdin deficient mice for three generations at the University of Leicester’s designated establishment. Mice were genotyped for the *Cfp* and *Fas* loci by PCR of genomic DNA, as described in [20,21]. The approval of the programme of work (Complement properdin in immunity and inflammation) was granted by the institutional Animal Welfare and Ethics Subcommittee (item AWERB/15/24) and by the Secretary of State of the UK Home Office (license P43308E3B). The severity of the protocol used in this paper was classed as moderate.

Mice were group housed (up to six mice) in a specific pathogen-free barrier facility in groups in ventilated cages at 21 °C, 50% humidity, with a 12/12 h light/dark cycle, and had ad libitum access to food and water. Mice were maintained on 5LF2 (TestDiet). The cage floor was covered with corn cob as bedding material; nesting material (sizzle pet) was made from recycled paper.

MRL/*lpr* mice develop lupus-like disease spontaneously. Skin vasculitis was not a feature of this *Fas* mutation but signs of lymphoproliferative disease (splenomegaly, lymphadenopathy) were manifest as of 4 months of age. This determined our humane endpoint. For this reason, no comment on the comparative longevity of the two genotypes can be made.

To increase the stringency of our comparisons, we decided to compare only male littermates in this study, because these would differ only in the presence or absence of properdin, due to x-chromosomal linkage of the properdin gene. The mice of interest to this study were produced by mating MRL/*lpr* properdin wildtype males with MRL/*lpr* properdin heterozygous females, which yields females which are MRL/*lpr* properdin wildtype or heterozygous and males which are either MRL/*lpr* properdin wildtype or MRL/*lpr* properdin hemizygous—i.e., deficient.

### 2.2. Histopathology

Microscopic analysis was performed blinded to the genotypes using Periodic Acid-Schiff (PAS) stained 5 micron sections. Immunoreaction with anti-mouse immunoglobulin HRP was detected using the chromogen 3,3′-diaminobenzidine.

### 2.3. Anti-DNA ELISA

Enzyme-linked immunosorbent assay (ELISA) was used to measure the levels of anti-DNA IgG autoantibodies in the serum samples. Deoxyribonucleic acid sodium salt from calf thymus (Sigma D1501) was added to poly-L-lysine coated plates [22].

### 2.4. C9 Functional Complements ELISAs

ELISA quantitatively measured the activity of three complement pathways at the level of C9 formation [23]. Activities were expressed as % of the activity of a commercial normal mouse serum (Thermo Fisher Scientific, Loughborough, UK). Experimental controls were serum heat inactivated at 56 °C for 30 min, or serum chelated by the addition of 5 mM EDTA.

### 2.5. BAFF and TNF-α ELISAs

Serum samples were diluted 1:100 and used for these analyses according to the manufacturer’s protocols: BAFF (Mouse BAFF/BLyS/TNFSF13B Quantikine ELISA Kit; R&D Systems, Abingdon, UK) and TNF-α (Murine TNF-α Mini TMB ELISA Development Kit; Peprotech EC Ltd., London, UK).

### 2.6. Serum Creatinine

This colorimetric assay was performed according to the manufacturer’s protocol (QuantiChrom^TM^ Creatinine Assay Kit; BioAssay Systems, Hayward, CA).

### 2.7. Caspase-3 Western Blot

Kidney lysates were prepared from experimental mice, separated by 12% SDS PAGE and electroblotted to nitrocellulose and probed with mouse Anti-Mouse Caspase-3 Antibody (Santa Cruz Biotechnology Inc, Heidelberg, Germany, Sc-56053; 1:400). After detection with Goat Anti-Mouse IgG H&L (HRP) (Abcam plc, Cambridge, UK, ab6789), blots were reacted with the HRP Chemiluminescent Substrate Reagent Kit (Novex^®^ ECL). The signal was captured using the ChemiDoc Imaging system (BIORAD). β-actin reactivity was used as a loading reference (Sigma Aldrich Co, Gillingham, UK, AC-74).

### 2.8. Statistical Analysis

Experimenters were blinded to the genotypes in all assays. Statistical values were determined by unpaired *t*-test. A *p* value < 0.05 was considered significant.

## 3. Results

### 3.1. Assessment of Renal Histopathology

Mice were analysed from 4 months of age prior to developing significant lymphadenopathy, splenomegaly and interstitial nephritis, which constituted our humane endpoint. It was hypothesised that a genotype specific contribution towards the progression of disease would be manifest. Histological analysis showed the epithelialization of Bowman’s capsules, possibly indicative of a need for greater protein absorption in both, MRL/*lpr* P^WT^ and MRL/*lpr* P^KO^ (Figure 1a). The degree of matrix expansion and cell infiltration was variable. IgG was localized to glomerular capillary endothelium when the sections were reacted with HRP-conjugated anti mouse immunoglobulins (Figure 1b).

To compare the extent and progression of glomerulonephritis, genotype matched pairs of one litter were humanely killed at three timepoints (68, 104, and 126 days), and 100 glomeruli for each genotype and timepoint were scored as focally, segmentally, or globally affected (Figure 2a). At 68 days, MRL/*lpr* P^WT^ and MRL/*lpr* P^KO^ showed a comparable distribution of glomerulonephritic changes. At 104 days, however, 56% of glomeruli in MRL/*lpr* P^WT^ were scored as global glomerulonephritis vs. 40% in MRL/*lpr* P^KO^ (Figure 2b). A clear distinction of histopathological phenotypes was seen at 126 days—MRL/*lpr* P^WT^ had even markedly more glomeruli with global inflammatory involvement than MRL/lpr P^KO^ (80% vs. 54%). In fact, the MRL/*lpr* P^KO^ mice exhibited a distribution of histological patterns at 126 days than compared to that of MRL/*lpr* P^WT^ at 104 days. This implies that MRL/*lpr* P^KO^ expressed a delayed disease phenotype.

### 3.2. Assessment of Disease Activity

Disease activity was assessed cumulatively at a humane endpoint of 4 months. MRL/*lpr* P^WT^ compared to MRL/*lpr* P^KO^ showed significantly increased levels of the metabolite creatinine (Table 1). Because creatinine elevation is an insensitive measure of renal impairment in mice [24], the doubling of levels in MRL/*lpr* P^WT^ compared to MRL/*lpr* P^KO^ and of MRL/*lpr* P^KO^ compared to the C57Bl/6 wildtype controls are pathologically important. The B cell activating factor belonging to the TNF family, BAFF, and cytokine TNFα were significantly increased in the sera of MRL/*lpr* P^WT^ compared to MRL/*lpr* P^KO^ mice (Table 1).

BAFF aids in the survival of B cells and has become a therapeutic target in clinics [25] and TNFα has a role in disease progression in MRl/*lpr* [26]. MRL/*lpr* P^WT^ also showed more renal pro-caspase-3 activation compared to MRL/*lpr* P^KO^ mice (Figure 3a,b). Caspase 3 is likely in this context to direct the activation of cytokines [27]. The anti-DNA antibody titre was significantly increased in MRL/*lpr* P^WT^ compared to MRL/*lpr* P^KO^ mice (Figure 3c).

In SLE, there is the constant systemic activation of complement in vivo. Ex vivo activity tests therefore quantify a decline relative to the functional consumption in vivo of components that are necessary in the assays. Therefore, the consumption of immune complex mediated complement activation was assessed by pathway specific activity assays (Table 2). Note that residual activities are measured, except for the test of alternative pathway activity in MRL/*lpr* P^KO^. This is because the alternative pathway test is absolutely reliant on the presence of properdin, so no activity is detected in the absence of properdin.

While there is a clear consumption of classical pathway activity in MRL/*lpr* P^WT^, MRL/*lpr* P^KO^ retained more activity to activate the classical pathway on the plate coated with IgM. The absence of properdin removes a significant amplifying activity of the alternative pathway to ongoing classical pathway of complement activation. Therefore, this assay quantifies the contribution afforded by the intact alternative pathway loop to ongoing classical pathway activation. The fact that the lectin pathway was consumed may point to the fact that convertase formation via the classical pathway draws away from lectin pathway progression. A secondary lectin pathway deficiency ensues, not necessarily consumption, and this secondary deficiency was less, as expected, in alternative pathway deficient MRL/*lpr* P^KO^. The consumption of complement factors in MRL/*lpr* P^WT^ extended to the alternative pathway, which does not require C4 or C2, components which are usually depressed in human SLE. An interpretation might be that there is the sequestration of activity to local, glomerular deposits.

## 4. Discussion

We have shown previously that complement system activation, and, in particular, properdin amplified classical pathway activation, is pathogenically involved in immune complex diseases [28]. The present studies demonstrate a key role for properdin in the pathogenesis of autoimmune renal disease in MRL/*lpr* mice.

The disease mechanisms of SLE can be studied in mice using models of lymphoproliferation. So far, crosses of the lupus prone mouse line MRL/*lpr* with either complement factor B, factor D or MASP-1/3 knockout mice demonstrated a significant benefit in targeting the alternative pathway of complement activation to reduce the severity of the renal pathology initiated by inflammation caused by immune complexes trapped in glomerular capillaries [17,18,29]. Properdin is the only positive regulator of the alternative pathway and its blockade is of clinical interest [30]. We showed that, in the absence of properdin, glomerulonephritis was less severe and progressed more slowly—disease activity markers were lower at the 4 month endpoint, which included autoantibody levels, inflammatory cytokines, complement activation, and tissue damage (Figure 4).

The two key observations comparing MRL/*lpr* P^WT^ and MRL/*lpr* P^KO^ were: (i) worse or quicker kidney inflammation, due to greater immune complex-mediated complement activation in MRL/*lpr* P^WT^, and (ii) more autoantibodies due to greater B cell activation in MRL/*lpr* P^WT^. The observed, ameliorated, phenotype in MRL/*lpr* P^KO^ compared with congenic MRL/*lpr* P^WT^ cannot be explained by just the absence of alternative pathway amplification caused by immune complex mediated complement activation. The reason for lower autoantibody production in MRL/*lpr* P^KO^ is not clear, but, based on current developments in the field [31,32], likely involves relative changes in signalling strengths via CR2/C3d, which may be exploitable for clinical application [33]. CR2 is of interest because CR2 is part of the B cell receptor complex and C3d binding leads to augmented BCR signalling.

Properdin deficiency causes a reduction in—not absence of—complement activation products, so the C3d/CR2 axis—and also the C5a/C5aR axis (relevant for B cell activity and egress [34])—is likely to be differentially engaged in MRL/*lpr* P^KO^ vs. MRL/*lpr* P^WT^. A measurable effect of properdin deficiency on B cell expression or activity of CD21 would strengthen the rationale for targeting properdin—or indeed of the ligand C3d [31,35]—in the treatment of lupus (while C5aR blockade has been shown to alleviate blood–brain barrier leakage in lupus prone mice [36]).

In addition, a report on the production of BAFF by renal tubular epithelial cells [37] raised the possibility that the kidney itself contributes to the progression of SLE via BAFF production and the retention of lymphocytes. This contribution could be less when properdin is targeted because glomerulonephritis is of delayed severity in MRL/*lpr* P^KO^. BAFF is a biomarker measured in patients with lupus. Anti-BAFF treatment (belimumab), however, has had variable success.

Recapitulating the beneficial effect of properdin deficiency for lupus glomerulonephritis using a validated anti mouse properdin antibody [38] would provide proof-of-principle that serum properdin is the dominant factor in determining disease activity and lupus phenotype. This seems likely as our work using MRL/*lpr* P^het^ mice (not shown) revealed a gene dose effect in alternative pathway activity but not in anti-dsDNA levels, IL-6 or TNFα levels—MRL/*lpr* P^het^ coincided in these measurements with MRL/*lpr* P^KO^, meaning that one of two copies of the properdin gene, although it reduced alternative pathway activity, was insufficient at driving a worse phenotype, as observed in MRL/*lpr* P^WT^. Our study supports previous work by others who targeted Factor B or D of the alternative pathway in their analyses of MRL/*lpr* mice and found that disease parameters, such as proteinuria or serum creatinine, significantly lessened at more prolonged endpoints [17,18]. While the absence of Factors B or D abrogates the assembly of the convertase complexes C3bBb and C3b_n_Bb (Factor D cleaves Factor B to produce Bb), the absence of properdin leaves the enzyme complexes labile to decay. The therapeutic targeting of properdin in lupus nephritis would likely require vaccinations against meningococcal disease [39]. Previously, the therapeutic application of a soluble form of a complement receptor of the Immunoglobulin superfamily, which functions as an inhibitor of the alternative pathway, led to a reduction in lupus nephritis at a comparable endpoint (4 months) in MRL/*lpr* mice, but, in contrast to our study, showed no effect on the levels of autoantibodies [40].

## 5. Conclusions

Our in vivo study shows the significant role of complement properdin in the disease activity of systemic lupus-like disease and in the severity of lupus nephritis.

There are significantly fewer anti DNA antibodies in serum from MRL/*lpr* mice in the absence of properdin. The difference in disease progression was deduced from histopathologic analysis of littermates, which crudely assesses severity. Relevant markers derange much earlier and their analyses in situ would help in determining which properdin supported inflammatory axis would benefit from other clinical blockers. Additional proteomic analyses could identify other ultimately targetable disease descriptors that associate with delayed disease progression in the absence of properdin.

## Figures and Tables

**Figure 1 medicina-56-00430-f001:**
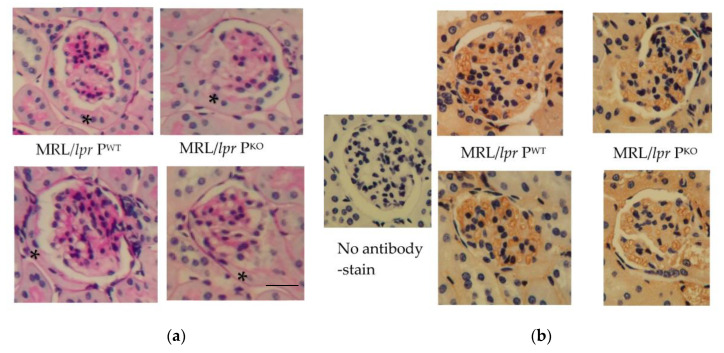
Representative images of glomeruli after staining with PAS (**a**) and after reaction with anti-mouse immunoglobulin G-HRP (**b**) *n* = 2 each genotype, MRL/*lpr* P^WT^ and MRL/*lpr* P^KO^. * indicates epithelialization of Bowman’s capsule. scale bar, 20 micron.

**Figure 2 medicina-56-00430-f002:**
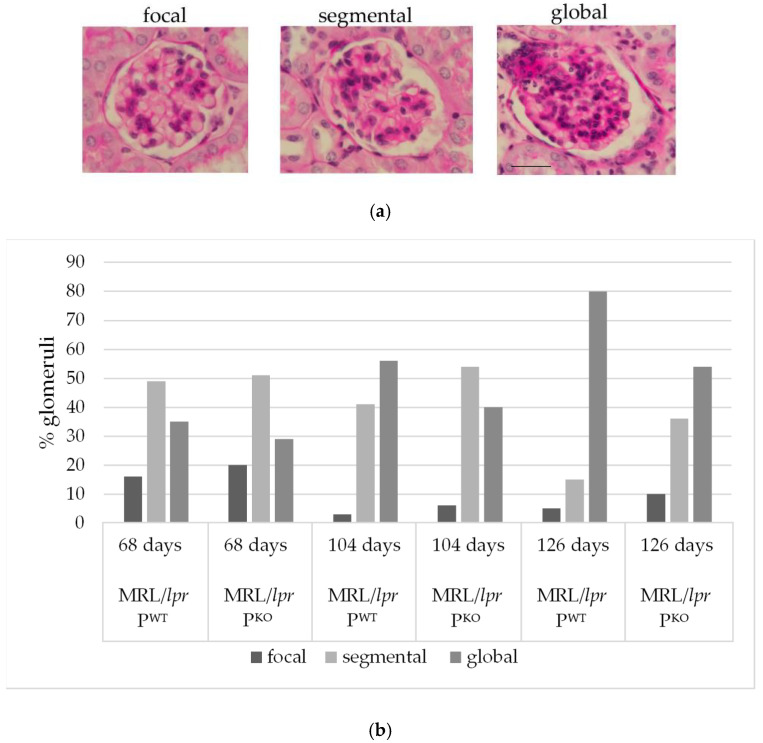
Examples of scored glomerular changes; scale bar, 20 micron (**a**). Histological evaluation of the extent of glomerulonephritis in 100 glomeruli of paired male littermates humanely killed at different times, as indicated (68, 104, and 126 days) (**b**).

**Figure 3 medicina-56-00430-f003:**
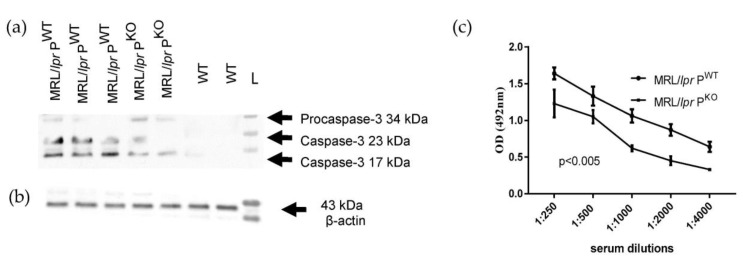
Western blot of kidney lysates prepared from MRL/*lpr* P^WT/KO^ and wildtype mice, probed for caspase-3 reactivity (**a**) and β-actin (**b**). (**c**) Anti-DNA Immunoglobulin G in sera of MRL/*lpr* P^WT^ and MRL/*lpr* P^KO^ (*n* = 3 each genotype).

**Figure 4 medicina-56-00430-f004:**
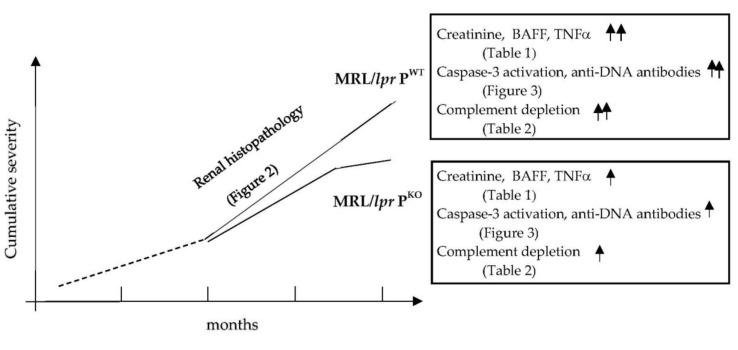
Summation of findings achieved in this study of MRL/*lpr* P^WT^ and MRL/*lpr* P^KO^.

**Table 1 medicina-56-00430-t001:** Measurements of creatinine, BAFF, and TNFα in serum.

	MRL/*lpr* P^WT^ *n* = 8	MRL/*lpr* P^KO^ *n* = 7	Unpaired *t*-Test (MRL/*lpr* P^WT^ cp MRL/*lpr* P^KO^)
Creatinine (mg/dL) (baseline (4 WT)): 0.29 ± 0.05	0.82 ± 0.08	0.49 ± 0.03	*p* < 0.0001
BAFF (pg/mL) (baseline (4 WT)): 31.17 ± 2.32)	70.5 ± 3.42	55.71 ±3.73	*p* < 0.0001
TNFα (pg/mL) (baseline (4 WT)): 7.1 ± 1.8	43.05 ± 3.21	18.46 ± 2.77	*p* < 0.0001

**Table 2 medicina-56-00430-t002:** Pathway specific complement activity pooled samples (*n* = 3) expressed as % activity of commercial normal mouse serum (set at 100%).

	MRL/*lpr* P^WT^	MRL/*lpr* P^KO^
Classical Pathway (Purified IgM)	48%	66%
Lectin Pathway (Mannan from *Saccharomyces cerevisiae*)	37%	48%
Alternative Pathway (*Salmonella enteritidis* LPS)	25%	1%

LPS, lipopolysaccharide.

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
