# Peer review of "Complement Properdin Determines Disease Activity in MRL/lpr Mice"

_medicina, 2020, doi:10.3390/medicina56090430_

Round 1

Reviewer 1 Report

This study used a new mouse model produced by crossing upus-prone MRL/MpJ-Faslpr/J mice with properdin knock-out mice to generate lupus-prone strain of properdin-deficient (MRL/lpr PKO) mice, and lupus-prone properdin-sufficient littermates (MRL/lpr PWT) as control. The authors of this study hypothesized that in the genetic absence of properdin, the systemic and organ specific inflammation might be less severe. The study found significantly reduced pathological and functional renal disease in MRL/lpr PKO mice compared to the wild type control showing that properdin and therefore alternate pathway plays a significant role in the proliferative glomerulonephritis that develops in MRL/lpr mice.

While interesting, this study has some major concers.

The authors have plagiarized themselves in the Abstract section. The submitted manuscript has been copied and pasted almost verbatim from a thesis that has been published online

The authors need to rewrite the abstract entirely to make it look different.

Also, kindly ensure that none of the other sections have materials directly copied from the thesis.

Line 13, page 2: The authors state that SLE affects 1 in 2000 Europeans. This could lead some readers to assume that it only affects Europeans. Therefore, please provide rate at which it affects African-Americans and others.

Line 13, page 2: It is not clear here in which condition “Antinuclear antibodies are produced”, are the authors meaning SLE in general or in the animal model studied?

Also, the statement “immune complexes may be deposited in any tissue causing vasculitis, arthritis, encephalitis, fetal loss, and nephritis” needs to be rewritten to remove any misgivings that these immune complexes can cause any of these conditions in any tissue. E.g deposition in joints cannot cause nephritis, etc.

Line 43, page 2- need to expand AP when first used.

Also, need to expand CP when first used

The authors need to discuss more about properdin, since this manuscript is about this protein. Which cells secrete properdin, where it can be found, some information regarding properdin deficiency in humans, etc?

Methods section

There is no evidence of Institutional approval for carrying out animal studies—a serious omission

The authors need to provide an experimental set up stating the number of animals used in the experiment, number of cages per animal, sex of animal used, etc. It would be good to have a schematic for the timeline of the experiment.

Anti-DNA assay

Deoxyribonucleic acid sodium salt from calf thymus (Sigma D1501) was used to coat the plates. Does the DNA coat the plate by itself or did the authors use protamine sulfate or polylysine first to coat to the plate followed by adding DNA to bind to protamine sulfate or polylysine?

The authors need to show that the sera of properdin knock-out mice did not have properdin as compared to wild type mice. Perhaps they can also show immunohistochemistry of properdin in tissues.

Is there data for properdin levels in human SLE flare and quiescent SLE? Is there data showing that patients with higher properdin levels develop progressive end stage renal disease?

The authors need to discuss the implication of this study for treatment of human SLE.

It would be useful to discuss the life span of properdin wild type versus properdin KO mice.

Minor comments

In some places the authors write Table 1, some places as table 1. Similarly, for fig 1 and Fig

Need to pay attention to spacing between words

Author Response

Thank you for the comments. The abstract has been rewritten, other parts worked over, and reference to the PhD thesis has explicitly been made. The points of critique that pertained to all sections of the manuscript have been dealt with and are all highlighted in yellow. Information on levels of properdin in humans has been included, and the implications for human disease management expanded. Our characterisation of the properdin deficient state has been made clearer.

Reviewer 2 Report

In this manuscript, the authors evaluated the role of properdin in a lupus-like mouse model (MRL/Lpr). They mainly evaluate the effect of this protein in the development and progression of glomerulonephritis but also some other markers including blood creatinine, complement activation, and BAFF/TNF quantities. Although this study seems to confirm the involvement of properdin in the lupus progression, the main effect is observed during the late steps of the pathology (126 days), and this could be better discussed. Also, the discussion could be improved with the description of different properdin inhibitors (antibodies, others...) that could be used in this study or the ongoing ones.

The introduction of the role of properdin in complement activation/regulation has to be improved.

How do the authors explain that the observed differences between Ko and wild type mice on glomerulonephritis occur only at later time points (126 days)? Could the authors describe the applied statistics for this experiment (Fig.2)?

Please the authors have to better describe how they obtain the MRL/lpr PKo mice (how many back-crossing with the MRL strain, male/female used for the study, how many mice are used for each experiment?)

Statistic analyses have to be better described.

Author Response

Thank you for the comments. We have improved the description of the time course, included possible experimental avenues, and expanded on the role of properdin. The crossing of mice to obtain experimental animals is now included. Thoughts on possible disease mechanisms have been included. All changes are highlighted in yellow.     

Round 2

Reviewer 1 Report

The authors have addressed most of the concerns of this reviewer.

Just some minor points-

The authors now state:

Mice were group housed in a specific pathogen-free barrier facility in groups in ventilated cages 25 at 21°C, 50% humidity, with 12/12 h light/dark cycle, and had ad libitum access to food and water.

It would be useful to state the number of animals per group in each cage.

Also, it will be useful to have a cartoon figure explaining the timeline of the experiment and showing timepoints where different pathologies appear in the mice.

Author Response

Thank you for the comments, the points have been addressed in M&M and a summative figure has been added in Disc. Please see green highlight for quick reference. 

Reviewer 2 Report

The authors address most of my concerns... I still have a question around their heterozygous properdin mice since this is an X-linked gene and they use males?

Author Response

The mating strategy has been expanded, please see green highlight for quick reference. Thank you.